# Early Bird Scheme for Parking Management: How Does Parking Play a Role in the Morning Commute Problem

Zipeng Zhang * 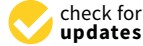 and Ning Zhang

School of Economics and Management, Beihang University, Beijing 100191, China; nzhang@buaa.edu.cn
* Correspondence: zipengzhang@buaa.edu.cn

**Abstract:** This paper extended the Vickrey's point-queue model to study the early bird parking mechanism during morning commute peak hours. We not only investigated how commuters choose departure times in view of morning commute traffic congestion and the discounted early bird parking fee, but also analyzed the conditions which are determined for the existence of the user equilibrium in the analysis model provided in this paper. Moreover, the tendency of the total queuing time and the incremental parking pricing revenue was derived along with the different choice strategy between early bird parkers (ERPs) and regular parkers (RPs). The results showed that the number of commuters was jointly determined by the desired time and the bottleneck capacity for different schedules. Additionally, the method of fare incentive showed a better effect on reducing queue than the initial no-incentive method with the instantaneous travel demand. Most importantly, the incremental parking revenue can be increased by properly adjusting the parking pricing gap between ERPs and RPs. Our research not only provided several important propositions for the early bird parking mechanism but also included the optimal solutions for optimal parking pricing and schedule gap in two groups of parkers. This work is expected to promote the development of early bird parking to mitigate morning commute traffic congestion and motivate the related research of schedule coordination for regulating parking choice behavior in morning peak hours.

**Keywords:** departure time choice; bottleneck model; early bird parking; morning commuting problem

## 1. Introduction

Urban parking management (UPM) has emerged as a very complicated and intractable problem in recent years following a renewed interest in rapid urbanism and growing environmental pollution, terrible congestion, and economic concerns, especially with the development of new mobility services and technologies, which bring the convenience and efficiency of parking and transportation demand management while increasing the imbalance between supply and demand, especially in the morning peak hours. This, consequently, entails the necessity to study the economic principle of parking management in order to direct city managers to figure out how to arrange and control their parking system more efficiently.

Recently, UPM problems have been regarded as a timely issue [1–3], which is undergoing a paradigm shift from the supply paradigm to the management paradigm (details are shown in Table 1). The supply paradigm of UPM [4] assumes that parking space supply should generally preferably be free and plentiful at most destinations. Some research [5,6] found that an ample parking supply and inexpensive parking price forces commuters to pay more and drive more than they otherwise need to. Moreover, the inefficient supply paradigm of UPM can also increase traffic congestion problems, especially in morning commute peak hours. Otherwise, the urban parking affordability (e.g., Beijing in China, which holds a mono-centric spatial structure) becomes more serious under the excessive and inflexible parking space requirements. Except for the limitation of land supply, Ref. [7]

indicated that it also requires more constant mechanical ventilation if the structured and underground parking infrastructures are less than 50% open, which is a continuing cost, plus security and fire suppression and the operating costs for maintenance. Accordingly, it is essentially preposterous to balance the abundant parking space and lower-priced urban parking service. It is worth noting, however, that it might be supportable for the parking solution of the supply paradigm in the suburbs where land space is generous and land value is economical. The solutions of the management paradigm tend to be better than the expanding supply in big cities, which hold a costly value of land and have a perfect public transport network. The new paradigm for the parking solution in mono-centric cities which favor a reduced parking supply more than a sharing strategy of parking facilities, more efficient regulations and pricing policies, and incentives in public transit modes has become a hot research direction of UPM. It includes strategies such as improved user information [8], more convenient payment systems, and improved travel options. More efficient parking management can benefit everybody, including motorists, businesses, and residents. UPM contradicts many community goals, including reducing traffic problems, increasing fairness and affordability, and more design flexibility.

**Table 1.** The definition and solutions of UPM problems.

|  | Supply Paradigm | Management Paradigm |
|---|---|---|
| Reason of UPM problem | Inadequate parking supply | Imbalance between supply and demand; unjustified pricing strategy; asymmetrical parking information |
| Attitude of parking supply | Sufficient parking supply | Bayesian paradox in parking supply |
| Attitude of parking fees | Free or inexpensive parking fees | Reasonable fees as one of congestion control |
| Attitude of parking priority | First-come principle | Higher priority conducive to efficiency improvement |
| Attitude of parking requirements | Strict application | Flexible |
| Attitude of parking innovation | Under-proven and widely accepted | Encouraged |
| Attitude of parking management | Allowed only when increasing supply is infeasible | Widely applied as a primary role in UPM |
| Land-use strategy | Acceptable or even desirable | Automobile-oriented development is undesirable |

Based on the research of our previous study [9] for Vickrey's bottleneck model, we found that the congestion condition during morning peak hours not only can be affected by the commuting demand, but also restricted by parking management. Inspired by the previous bi-arrival bottleneck model [9], we focused on the research of management solutions such as the early bird parking strategy to optimize the traffic cost for morning commutes under the consideration of congestion and parking. The novel bottleneck model based on the early bird parking mechanism introduced in this paper includes staggering peak parking and differentiated charges; it can also be used to explore the effect on reducing traffic congestion and increasing the parking revenue. Most importantly, the exploration of parking revenue was the biggest contribution in this paper when compared with the previous model [9]. In this paper, we also answered the questions as stated below:

- How can the early bird parking strategy result in a more efficient use of parking resources?
- How can the early bird parking strategy reduce traffic congestion?
- How can the early bird parking strategy increase the revenue of parking management?

The rest of this paper is organized as follows. Section 2 displays the review of the literature in parking management. Section 3 describes the bi-arrival bottleneck model and the cost formulations for two group of commuters: early bird parker and regular parker in different schedule gap cases. In Section 4, the dynamic single-peak and double-peak queue traffic patterns of user equilibrium with mixed travelers in different pickup-work time interval situation are discussed, and the evolution of the dynamic queue over time is

also shown by analytic solution. Section 4 examines and analyzes the morning commute performance under the three strategies to achieve the objective of balancing the over-centralized demand. Numerical illustrations and verifications are presented in Section 5. Finally, we provide some conclusions in Section 6.

## 2. Review of the Literature

The abovementioned research objectives it entail the need to understand how to define the effect of parking congestion, what factors drive the utility of the parking system, the impact that existing the measure of early-bird parking have had on the efficiency of parking or the total travel system, and, finally what statistical mechanism of early-bird parking we can learn for achieving the aims of smoothing the travel demand during peak-hour period.

### 2.1. Measuring Traffic Congestion: Vickrey's Bottleneck Mode

Parking constraint plays a key role in the leading solutions to mitigate the travel congestion during the morning commuting period [10]. The analyzing bottleneck model for morning commute problem, as proposed by [11] and systematized especially by [12] (hereafter ADL model),is disputably the most fundamental advancement in the field of analysis and modeling of traffic congestion.

Some research [13,14] embedded a parking bottleneck based on the properties of Vickrey's model [11], and the analytical solution showed that charging for these on-street parking patterns can effectively reduce total travel congestion, and the scheme of parking-charging in urban areas can be regarded as a more acceptable policy relative to a congestion toll. Ref. [15] extended the on-street parking bottleneck model above to allow for both on-street and off-street parking. Ref. [16] established a bi-comparative model in one university campus to evaluate travelers' attitude for congestion tolls and parking fees; the results of SP survey indicated that the regulatory efficacy of congestion tolls in smoothing peak-hour travel demand is higher than parking fees, because drivers prefer paying for parking fees rather than congestion tolls when these two policies were implemented in parallel. Furthermore, Refs. [14,17] drew a conclusion that such a time-varying parking pricing, when constricted to be strictly increasing in arrival time, can shorten queuing length and minimize its duration. Moreover, under the on-street parking scheme developed by [13], Ref. [18] developed an extended bottleneck model to analyze the congestion phenomenon in the morning commuting-parking problem, and they investigated the optimal parking supply and congestion tolling schemes to minimize total travel cost. Similar analysis models were also developed later to explored the travelers' choice behavior in the daily commuting paradigm when morning and evening trip chains are completed under the mobility service environment of autonomous vehicles [19].

### 2.2. Parking Management Problem of Different Optimal Perspective

In this section, we display the previous researching work related to UPM problem of different optimal perspectives. Considering that traffic congestion is becoming a prevalent phenomenon with the development of urbanization, many researchers combine the congestion and parking as the studied topics in the economics of parking [20–22], which are broadly classified into three optimal perspectives: parking cruising, parking pricing, and the parking spaces constraint.

The cruising time for searching an available parking lot, regarded as a major contributing factor of emissions and congestion in the morning peak-hour period, often plays an important role in the choice of travel behavior. Refs. [23,24] introduced a stochastic traffic assignment model to design a choice pattern how drivers choose between cruising for parking or paying for parking when the spaces of on-street parking are not available immediately, but off-street parking is not all occupied at present but more expensive; the result predicted that it will eliminate cruising time until the toll of on-street parking patterns at least equal to the marginal cost of neighboring off-street ones. A significant amount of original studies on the cruising for parking problem (on-street or off-street parking pattern)

integrating traffic congestion were carried out by Arnott and his collaborators [10,13,15,25]. Based on the work above, Ref. [26] introduced an approach of mathematical programs with equilibrium constraints to explore the optimal location of on-street parking spaces in urban areas. Ref. [27] expanded the parking cruising model to an aggregated-network traffic model. By using the macroscopic fundamental diagram (MFD) theory, they achieved the time-based optimal equilibrium result. Ref. [28] established a dynamic programming model for UPM and in this model, and they reconciled the relationship between cruising for parking and time cost and found a self-regulating mechanism in the process of cruising; they also directed that varying of travel demand attributable to cruising is minimal when compared with parking fees from large-scale GPS data and a SP survey for households.

Besides the cruising for parking problem, parking pricing policies are also used in eliminating traffic congestion. These studies mainly focus on the parking pricing policy implementation in view of travel demand [16,29,30], travel choice behavior [31,32], and dynamic traffic flows [33,34]. In a static context, the research [35] verified the phenomenon that congestion can be caused by traffic destined for the parking area, and introduced parking fees as an essential factor to optimize the total travel time; numerical simulation showed that parking fees can increase social welfare when applying the second-optimal toll strategy, while an efficient time-varying toll may eliminate queuing to achieve the maximum social welfare. Ref. [36] built an e-Tag traffic management model to moderate urban congestion, and demonstrated how the replaceable parking fees can mitigate travel congestion. More recently, some studies [8,37] examined the sensitivity analysis of parking pricing policies to properties of the commuting system when it achieved user equilibrium to help shorten congestion time and cut commuting costs. Ref. [38] proposed two hybrid management schemes considering both optimal pricing and permission in a many-to-one mono-centric urban network; numerical simulation indicated that these hybrid parking schemes were more effective than another pure one of two schemes.

In addition to cruising for parking and parking pricing policies, many studies also focused on how the constraint of parking spaces shapes travelers' commuting behavior (e.g., choice of travel model and departure time). Almost all studies suppose that parking availability affects commuters' spatial and temporal departure choice, which result in morning commuters preferring to depart earlier to obtain an available parking lot when parking supply is deficient. Considering two parameters of departure time choice, likely schedule delays and parking constraint, Ref. [39] analyzed the effect of reserved parking scheme on the commuting behavior through Vickrey's bottleneck model, and analytical solution indicated that an optimal allocation proportion for reserved parking space can temporally relieve the total travel cost in commuting system. Ref. [40] expanded Yang's parking model to a novel tradable parking permit scheme under the policy of parking reservation among heterogeneous commuters. Otherwise, the travel behavior under parking space constraints considering carpooling mode, Ref. [41], and autonomous vehicle mode, Ref. [42], have been of particular interest to researchers.

Based on varying network specifications, there other various aspects associated with UPM problem, which are introduced in some literature, e.g., the effect of parking supply in multiple-mode networks [43]; parking reservation and pricing [44–48]; parking supply and ride-sourcing service [49–51]; parking space constraint and autonomous vehicles [19,52,53].

## 3. Model Framework

In this section, we focus on introducing the early bird parking model during the morning rush hours in a single corridor network.

### 3.1. Notations

The notations used in this paper are list as follows.

Model parameters (all positive scalars)

| | |
|---|---|
| $\alpha$ | Value of time |
| $\beta$ | Unit cost of early arrival penalty |
| $\gamma$ | Unit cost of late arrival penalty |
| $\tau_1$ | Parking fee of EBPs |
| $\tau_2$ | Parking fee of RPs |
| $\omega$ | Penetration rate of early bird parking commuters |
| $t_1^*$ | Desired working time |
| $t_2^*$ | Desired pickup time |
| $\Delta t$ | The schedule gap between early bird parkers and regular parkers |
| $\Delta \tau$ | The parking pricing gap between early bird parkers and regular parkers |
| $s$ | Capacity of the bottleneck (veh/h) |
| $N$ | Total commuting demand |

Time-varying variables

| | |
|---|---|
| $q(t)$ | Queue length at the bottleneck at time t |
| $T(t)$ | The total travel time for commuters departing at time $t$ |
| $T^w(t)$ | Queuing time in bottleneck departing at time $t$ |
| $r_1(t)$ | The equilibrium departure rate of EBPs |
| $r_2(t)$ | The equilibrium departure rate of RPs early for work |
| $r_3(t)$ | The equilibrium departure rate of RPs late for work |
| $c_1(t)$ | The travel cost of EBPs departing from home at time $t$ |
| $c_2(t)$ | The travel cost of RPs departing from home at time $t$ |

Intermediate notations

| | |
|---|---|
| $t_1^a, t_2^a$ | The earliest departure time for EBPs and RPs, respectively |
| $t_1^b, t_2^b$ | The latest departure time for EBPs and RPs, respectively |
| $\widehat{t_1}$ | The punctual departure time of RPs for work |
| $N_1, N_2$ | Travel demand for EBPs and RPs, respectively |

### 3.2. Model Description and Mainly Assumption

Considering a bottleneck-constrained corridor that connects a residential area and a central business district (CBD), during the morning peak hours, two groups of *N* continuum homogeneous commuters available on the corridor are early-bird drivers (EBDs), $N_1$, and standard work drivers (SWDs), $N_2$. Meanwhile, we assume that all parking spaces are located at the CBD. The early bird parking service, as a cheaper parking option, can be provided in all parks before desired first bird parking time ($t_1^*$) and the desired arrival times at the workplace for both kinds of commuters are assumed to be identical and equal to $t_2^*$. Let $t_0$, $t_0'$ be the free-flow time before and after reaching the bottleneck. In addition, we make the following assumptions:

(i)   Without loss of generality, assuming $t_0 = 0$, $t_0' = 0$, we can regard the length of bottleneck as the distances of OD pair in the single corridor; meanwhile, the departure time from the origin is equal to the arrival time at bottleneck, and the exiting time of the bottleneck is equivalent to the arrival time at CBD.

(ii)  The walking time from parking spaces to workplace and the cruise time in park is ignored without loss of generality. Traffic departure and arrival take place over the interval $t \in [t_s, t_e]$. According to Assumption (i), $t_s$, $t_e$ is also the earliest and the last time for commute entering the bottleneck, respectively.

(iii) Assume that $\omega$ is the penetration rate of RD commuters, then $N_2 = \omega N$ is the number of RD commuters; $N_1 = (1 - \omega)N$ is the number of SD commuters, and $N = N_1 + N_2$ always holds. Notably, $\omega = 0$ and $\omega = 1$ denote two extreme patterns in which the mixed-commute system includes only STDs or only EBDs, respectively.

### 3.3. Estimating Queuing Properties: Vickrey's Point-Queue Delay Model

The Vickrey's point-queue model, a straightforward approach to model traffic dynamics, can describe the formation and dissipation of queuing delay behavior during the morning peak hours.

Traffic forms a point-queue before entering the bottleneck when the commuters' departure rate $r_i(t)$ from home or arrival rate at bottleneck exceeds the lane's capacity (denoted as $c(t)$) during the interval $t \in \left[ t_i^a, t_i^b \right]$. Letting $A_i(t)$, $D_i(t)$ be the cumulative count curve of commutes departure from origin and arrival at destination at time $t$, respectively, the queue length at time $t$ is obtained by the following formula:

$$q_i(t) = D_i(t) - A_i(t) = \int_t^{t + T_i(t)} c(x) \, dx \tag{1}$$

$$q_i(t) = \begin{cases} \int_{t_i^a}^t (r(x) - c(x))dx & \text{if } q_i(x) > 0 \\ 0 & \text{otherwise} \end{cases} \tag{2}$$

The right hand side of Equation (1) indicates that the exit flow rate $c(\cdot)$ is a continuous-time dynamic function, which is assumed herein to be equal to the fixed bottleneck capacity $s$, (ignoring the effect about capacity drop phenomenon), because the dynamic one will be considered in our future research. Where $T_i(t)$ is the total travel time between OD pair, it can be regarded as the queuing time in bottleneck, $T_i^w(t)$, under Assumption (i). The change rate of the queue length can be simplified as an ordinary differential equation:

$$\frac{dq_i(t)}{dt} = r(t - t_i^a) - \begin{cases} s & \text{if } q_i(t) > 0 \\ \min\{r(t - t_i^a), s\} & \text{otherwise} \end{cases} \tag{3}$$

According to Equations (1)–(3) and Assumption (i), the total travel time $T_i(t)$ for commuters departing at time $t$ is:

$$T_i(t) = T_i^w(t) = \frac{q_i(t)}{s} \tag{4}$$

### 3.4. Estimating Trip Cost: Vickrey's Continuous-Time Schedule Penalty Model

According to the classical morning commute ADL model, the generalized travel cost, $c_i(t)$, for different groups of commutes departing at departure time $t$ can be formulated in this subsection.

For a SWD, the travel cost is the summation of the travel time cost, schedule delay penalty for work, and standard parking fare cost, while for FBDs, the extra travel cost includes the schedule early penalty for early bird parking and discount fare of parking. Let $c_1(t)$, $c_2(t)$ serve as the travel cost for EBDs and SWDs who depart at time $t$, respectively. We have:

$$c_1(t) = \alpha T(t) + \beta \cdot (t_1^* - t - T(t)) + \beta \cdot (t_2^* - t - T(t)) + \tau_1 \tag{5}$$

$$c_2(t) = \alpha T(t) + \max\{\beta \cdot (t_2^* - t - T(t)), \gamma \cdot (t + T(t) - t_2^*)\} + \tau_2 \tag{6}$$

where $\alpha$ is the value of time, $\beta$, $\gamma$ is the unit schedule delay penalty separately for early arrival and the late arrival. Without loss of generality, we assume $\beta < \alpha < \gamma$. $\tau_1$, $\tau_2$ is the parking fare for FBDs and SWDs, respectively. Normally, in order to encourage early-bird parking as described in this research, $\tau_1$, $\tau_2$ yields $\tau_1 < \tau_2$ and $t_2^*$ is the desired work time for all commuters. Denoting the free-flow time as zero and neglecting the walking delay from park to workplace, we can see that the arrival time of parking for FBDs and SWDs is equal to the arrival time of CBD.

## 4. Benchmark Departure Patterns for Two Extreme Patterns

Commuters are normally assumed to choose departure times to make a trade-off between the queuing cost and the schedule-delay cost under individually optimal conditions, resulting in a dynamic user equilibrium (DUE). At user equilibrium, the travel cost for these three groups of commuters who depart from the origin at time $t$ should be equal to each

other; no one can unilaterally shift their departure time to obtain more utility. This means $dc(t)/dt = 0$. We can easily obtain the behavior properties for a different group when the system achieves equilibrium. The detail is shown in Appendix A, and we emphasize the departure rate for SWDs and FBDs in this section.

### 4.1. Pattern 1: Departure Scenarios of DUE with Only EBDs

We firstly consider the extreme case with $\omega = 1$ or $N_1 = N$, which is called as benchmark pattern 1 to distinguish it from the pattern of $\omega = 1$. As analyzed above, there are only early bird drivers including in the commuting corridor when $\omega = 1$. We make $t_1^a$, $t_1^b$ as the earliest and latest departure time for EBDs, respectively. Differentiating Equation (5) with respect to $t$ and setting to zero, $dc_1(t)/dt = 0$, the equilibrium departure rate from home for EBDs who arrive at the parking place before the desired arrival time $t_1^*$ can be given by:

$$r_1(t) = \frac{\alpha}{\alpha - 2\beta} \cdot s, \qquad t \in (t_1^a, t_1^b) \tag{7}$$

The earliest and latest departure time for EBDs are provided below. The departure scenarios when commuting system achieve user equilibrium are show in Figure 1.

$$t_1^a = t_1^* - \frac{N_1}{s}, \ t_1^b = t_1^* - \frac{2\beta}{\alpha}\frac{N_1}{s} \tag{8}$$

$$T(t) = \frac{2\beta}{\alpha - 2\beta}(t - t_1^a), \ t \in (t_1^a, t_1^b) \tag{9}$$

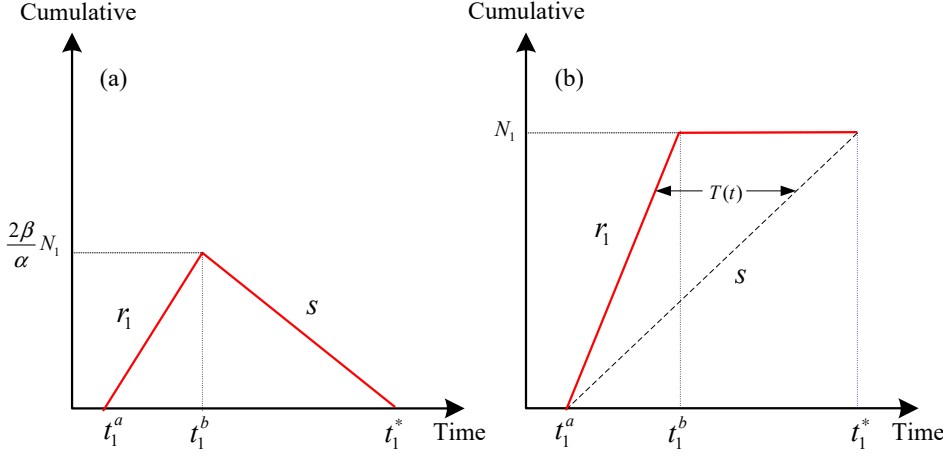

**Figure 1.** The queuing diagram and the equilibrium diagram with only EBDs: (**a**) Queuing length in the benchmark pattern 1; (**b**) Departure pattern of DUE in benchmark pattern 1.

At equilibrium, the individual travel cost from home to the workplace is

$$c_1(t) = 2\beta \cdot \frac{N_1}{s} + \beta \cdot \Delta t + \tau_1 \tag{10}$$

### 4.2. Pattern 2: Departure Scenarios of DUE with Only SWDs

Pattern 2 will occur when $\omega = 0$, in which only SWDs are driving on the commuting corridor. Let $t_2^a$ be the start time of the earliest RD commuters, $t_2^b$ be the departure time of the latest SWDs, and donate $\widehat{t}_2$ be the departure time of SWDs who arrive at CBD on time. Combining Equation (6) and differentiating Equation (6) with respect to $t$ and setting to zero, $dc_2(t)/dt = 0$, the equilibrium departure rates from home for work of SWDs are given by:

$$r_2(t) = \frac{\alpha}{\alpha - \beta}s \ t \in (t_2^a, \widehat{t}_2), \ r_3(t) = \frac{\alpha}{\alpha + \gamma}s \ t \in (\widehat{t}_2, t_i^b) \tag{11}$$

The earliest departure time, the departure time for on-time arrival at the destination, and the latest departure time are given by:

$$t_2^a = t_2^* - \frac{\gamma}{\beta + \gamma} \frac{N_2}{s}, \quad \widehat{t}_2 = t_2^* - \frac{\beta\gamma}{\alpha(\beta + \gamma)} \frac{N_2}{s}, \quad t_2^b = t_2^* + \frac{\beta}{\beta + \gamma} \frac{N_2}{s} \tag{12}$$

$$T_2^w(t) = \begin{cases} \frac{\beta}{\alpha - \beta}(t - t_2^a), & t \in (t_2^a, \widehat{t}_2) \\ \frac{\gamma}{\alpha + \gamma}(t_2^b - t), & t \in (\widehat{t}_2, t_2^b) \end{cases} \tag{13}$$

Similar to the description of Figure 1, the departure scenarios when commuting systems achieve user equilibrium are shown in Figure 2. At equilibrium, the individual travel cost from home to the workplace can be expressed as below:

$$c_2(t) = \frac{\beta\gamma}{\beta + \gamma} \cdot \frac{N_2}{s} + \tau_2 \tag{14}$$

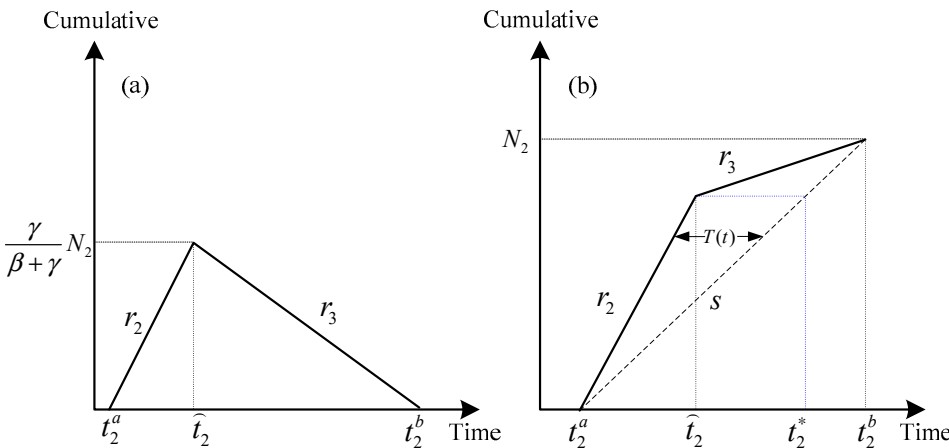

**Figure 2.** The queuing diagram and the equilibrium diagram with only SWDs: (**a**) Queuing length in the benchmark pattern 2; (**b**) Departure pattern of DUE in benchmark pattern 2.

## 5. Departure Scenarios with Mixed Commuters

Depending on the values of $N_1$, $N_2$, $\alpha$, $\beta$, $s$ and $\Delta t = t_2^* - t_1^*$, different equilibrium traffic patterns may arise. For instance, if $\Delta t = t_2^* - t_1^*$ is extremely large (much larger than $(N_1 + N_2)/s$), EBPs and RPs will travel at very different times and thus their travels will be completely separated; if $\Delta t = t_2^* - t_1^*$ approaches zero, one can expect that the two types of commuters would travel at similar times and thus interact with each other through sharing the same corridor. We present all possible equilibrium traffic patterns with mixed commuters in Figure 3. Conditions of the occurrence of each traffic pattern can be accordingly determined based on these time points. We summarize the conditions for the occurrence of each traffic pattern in Table 2.

Case 1 in Figure 3a: In this case, the schedule gap $\Delta t$ between early bird parking schedule and work is relatively large. All EBDs depart from home between $t_1^a$ and $t_1^b$, while RPs depart much closer to the work start time $t_2^*$ as they do not need to consider the early bird parking plan. The travels of EBDs and RPs are completely separated. However, corridor capacity is wasted between $t_1^*$ and $t_2^a$ when no one uses it.

Case (2) in Figure 3b: This pattern is similar to Pattern (1). The difference lies in that, as the schedule gap $\Delta t$ between EBPs and RPs becomes smaller, EBDs and RPs are now more connected, i.e., the first SWD will join the queue behind the last EBD. It follows that there is no capacity waste between the arrivals (at work) of EBDs and RPs.

Case (3) in Figure 3c: Case (2) is a critical case of this Case, two groups of commuters mix at the same time in the corridor between $t_2^a$ and $t_1^*$ or $t_1^*$ and $t_2^a$. We can not determine

the relationship between $t_2^a$ and $t_1^*$ yet. Using the expression in Equations (8) and (12), we deduce that, when $t_2^* - t_1^* < \frac{\gamma}{\beta+\gamma}\frac{N_2}{s}$, $t_2^a$ is no bigger than $t_1^*$, when $t_2^* - t_1^* < \frac{\gamma}{\beta+\gamma}\frac{N_2}{s}$, $t_2^a \geq t_1^*$ the detail of the relationship between $t_2^a$ and $t_1^*$ can be shown at Appendix A.

Case (4) in Figure 3d shows a special when the schedule gap $\Delta t$ between early bird special and regular work is equal to zero.

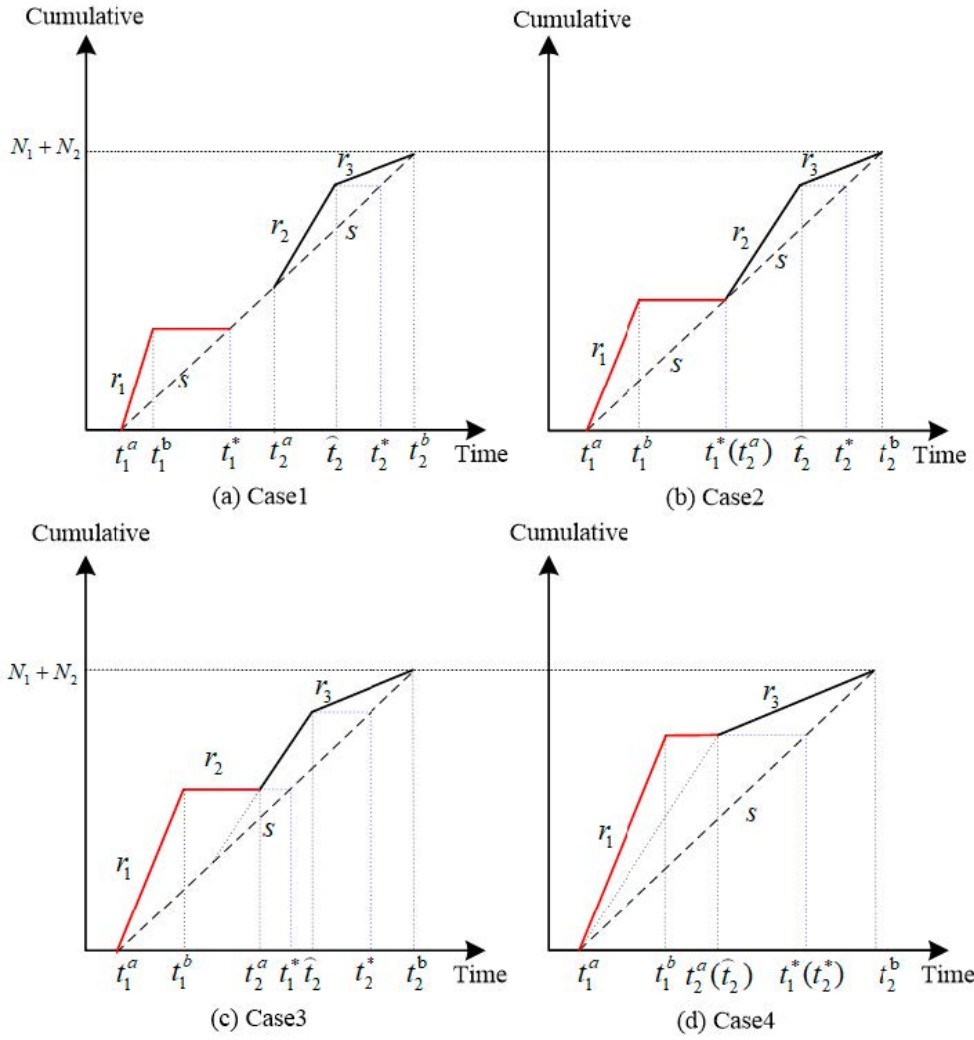

**Figure 3.** All queuing scenarios of two kinds of parkers: (**a**) when the schedule gap $\Delta t$ is relatively large; (**b**) when the schedule gap satisfies $\Delta t = \frac{\gamma}{\beta+\gamma}\frac{N_2}{s}$; (**c**) when the schedule gap satisfies $\Delta t < \frac{\gamma}{\beta+\gamma}\frac{N_2}{s}$; and (**d**) when there is no schedule gap.

**Table 2.** Conditions for the different scenarios depicted in Figure 3.

| Cases | Conditions | Figures | $t_1^a$ | $t_1^b$ | $t_2^a$ | $t_2^b$ |
|---|---|---|---|---|---|---|
| Case 1 | $\Delta t = t_2^* - t_1^* > \frac{\gamma}{\beta+\gamma}\frac{N_2}{s}$ | Figure 3a | $t_1^* - \frac{N_1}{s}$ | $t_1^* - \frac{2\beta}{\alpha}\frac{N_1}{s}$ | $t_2^* - \frac{\alpha}{\alpha-\beta}\frac{N_2}{s}$ | $t_2^* + \frac{\beta}{\beta+\gamma}\frac{N_2}{s}$ |
| Case 2 | $\Delta t = t_2^* - t_1^* = \frac{\gamma}{\beta+\gamma}\frac{N_2}{s}$ | Figure 3b | $t_1^* - \frac{N_1}{s}$ | $t_1^* - \frac{2\beta}{\alpha}\frac{N_1}{s}$ | $t_1^*$ | $t_1^* + \frac{N_2}{s}$ |
| Case 3 | $0 < t_2^* - t_1^* < \frac{\gamma}{\beta+\gamma}\frac{N_2}{s}$ | Figure 3c | $t_1^* - \frac{N_1}{s}$ | $t_1^* - \frac{2\beta}{\alpha}\frac{N_1}{s}$ | $\frac{\alpha-2\beta}{\beta(\beta+\gamma)}\frac{N_2}{s} - \frac{\alpha-\beta}{\beta}\frac{N_1}{s} + \frac{\alpha-\beta}{\beta}t_1^* - \frac{\alpha-2\beta}{\beta}t_2^*$ | $t_2^* + \frac{\beta}{\beta+\gamma}\frac{N_2}{s}$ |
| Case 4 | $\Delta t = t_2^* - t_1^* = 0$ | Figure 3d | $t_2^* - \frac{N_1}{s}$ | $t_2^* - \frac{2\beta}{\alpha}\frac{N_1}{s}$ | $\widehat{t}_2$ | $t_2^* + \frac{\beta}{\beta+\gamma}\frac{N_2}{s}$ |

### 5.1. Parking Schedule Gap Coordination

Now we explore how to coordinate the schedules of early bird parking and regular parking to reduce travel cost when the parking fee gap $\Delta\tau$ is fixed. The total travel cost for all commuters can be expressed as follows:

$$TTC = N_1 \cdot c_1 + N_2 \cdot c_2 \tag{15}$$

$$TTC = \begin{cases} (2\beta \cdot \frac{N_1}{s} + \beta \cdot \Delta t + \tau_1) \cdot N_1 + (\frac{\beta\gamma}{\beta+\gamma} \cdot \frac{N_2}{s} + \tau_2) \cdot N_2 \\ (2\beta \cdot \frac{N_1}{s} + \frac{\beta\gamma}{\beta+\gamma} \cdot \frac{N_2}{s} + \tau_1) \cdot N_1 + (\frac{\beta\gamma}{\beta+\gamma} \cdot \frac{N_2}{s} + \tau_2) \cdot N_2 \\ (2\beta \cdot \frac{N_1}{s} + \beta \cdot \Delta t + \tau_1) \cdot N_1 + (-\gamma\Delta t + \gamma\frac{N_2}{s} + 2\beta\gamma\frac{N_1}{s}) \cdot N_2 \\ 2\beta \cdot \frac{N_1}{s} + (\gamma\frac{N_2}{s} + \tau_2) \cdot N_2 \end{cases} \tag{16}$$

When the parking fee gap between early bird parkers and regular parkers is fixed, the optimal parking schedule gap can be determined, which is denoted by $\Delta t^*$.

According to the descriptions in Table 2 and Figure 3, $N^*$ can be determined (shown in Figure 3b), $N_1^* = N - N_2^*$, $N_2^* = \Delta t \cdot s \cdot (\beta + \gamma)/\gamma$ when the total travel demand is $N > N^*$, meanwhile $N_1^* = N - N_2^*$, $N_2^* = N_1 \cdot 2(\beta + \gamma)/\gamma$, when the total travel demand is $N < N^*$. The derivations of this value are shown in Appendices B and C.

### 5.2. Parking Fee Gap Coordination

The parking fee gap is $\Delta\tau = \tau_2 - \tau_1$, where $\tau_1$, $\tau_2$ is the parking fee of early parkers and regular parkers, respectively. For the given number of different groups, we can acquire the total parking benefit function of the parking manager:

$$R = N_1 \cdot \tau_1 + N_2 \cdot \tau_2 \tag{17}$$

In this paper, we suppose the total parking benefit $R_0 = 0$ for the no-early-bird parking management strategy, $\Delta\tau = 0$, and the early bird parking fee strategy can be regarded as $\Delta R = R - R_0$, which can be also regarded as $\Delta R = N_2\Delta\tau$, when $\tau_1 = 0$.

When parking demand is larger than standard parking schedule gap ($N > N^*$), according to Theorem 1, under the condition of user equilibrium in the system, we know that the number of early bird parkers and regular parkers is $N_1 = N - N_2$, $N_2 = \frac{\beta+\gamma}{\gamma} \cdot \Delta t \cdot s$. This means that the number of regular parkers is fixed when the parking schedule gap $\Delta t$ is fixed in the case of $N > N^*$; it needs more early bird parkers when the parking demand is large. When the system achieves user equilibrium when the condition of parking demand is less than standard parking schedule gap ($N < N^*$), we also can obtain the relationship between the number of early bird parkers and regular parkers: $N_2 = \frac{2(\beta+\gamma)}{\gamma} \cdot N_1 + \frac{\beta+\gamma}{\gamma} \cdot \Delta t \cdot s - \frac{\beta+\gamma}{\beta\gamma}\Delta\tau \cdot s$. Table 2 provides the detail of the total travel time and the increase in parking revenues in different cases of large and small demand, whose derivations are shown in Appendices B and C.

## 6. Numerical Analysis

In this section, we display numerical results to illustrate and verify the dynamic user equilibrium and system performance described above in Sections 4 and 5, which also prove four early bird parking management questions: (1) the effect of morning traffic congestion (bottleneck capacity) on morning commuters' parking choice behavior when the parking schedule gap and parking pricing gaps are fixed. (2) The effect of the parking schedule gap and bottleneck capacity on the allocation of early bird parkers and regular parkers when the commute demand is fixed. (3) The effect of the total commute demand on the total travel time and increases in parking revenue when parking schedule gap and bottleneck capacity is fixed. (4) How to optimize the parking pricing gap to achieve the objective of reducing congestion and increasing parking benefit.

In the simulation part, a synthesized real-life network (as shown in Figure 4) of a morning-commuting single corridor (in Shenzhen, China) was considered, which was

divided into six segments. The details about the network parameters and their calibration are showed in Table 3. We set parameters of the cost in value of time, with schedule delay penalties as follows: $\alpha$ = 2.0 (HK\$/min), $\beta$ = 1.0 (HK\$/min), $\gamma$ = 3.0 (HK\$/min), and s = 10 (veh/min).

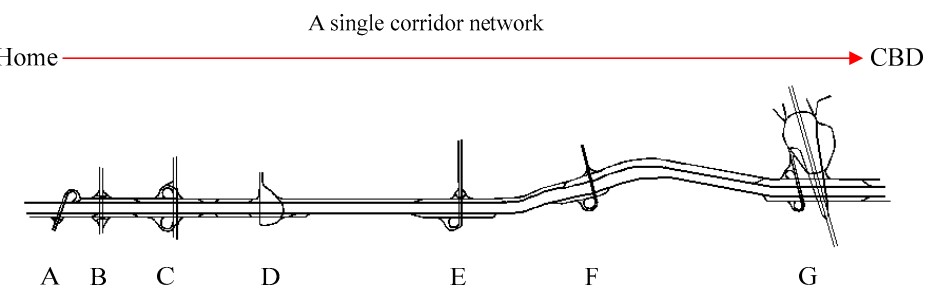

**Figure 4.** A single corridor network in Shenzhen City.

**Table 3.** Description of segments in the network.

| Segment NO. | Description | Length | Number of Lanes |
|---|---|---|---|
| 1 | Houhai Overpass (A) and Keyuan Overpass (B) | 458 m | 3 |
| 2 | Keyuan Overpass (B) and West Shahe Overpass (C) | 770 m | 3 |
| 3 | West Shahe Overpass (C) and East Shahe Overpass (D) | 995 m | 3 |
| 4 | East Shahe Overpass (D) and Shenwan Overpass (E) | 1560 m | 3 |
| 5 | Shenwan Overpass (E) and East Qiaocheng Overpass (F) | 1200 m | 3 |
| 6 | East Qiaocheng Overpass (F) and Zhuzilin Overpass (G) | 2200 m | 3 |

### 6.1. The Occurrence Domains of ($N_1$, $N_2$) When System Achieve UE in Different Cases

In this section, we discuss the effect of morning traffic congestion (bottleneck capacity) on morning commuters' parking choice behavior when the parking schedule gap and parking pricing gaps are fixed. Due to this objective, we exemplify the occurrence of ($N_1$, $N_2$) when the system achieves UE in different cases, as detailed in Table 4. With the fixed parking schedule gap $\Delta t$ and parking pricing gap $\Delta \tau$, Figure 5 clearly illustrates the domains of occurrence for all different cases shown in Table 5 with respect to different values of $\Delta t$ and $\Delta \tau$. The solid black line represents the optimal allocation of commute demand $N^*$ ($N^* = N_1^* + N_2^*$) among different cases, which can be indicated from Section 5. We know that $N_1^* = N - N_2^*$, $N_2^* = \frac{\beta + \gamma}{\gamma} \Delta t \cdot s$ when the total travel demand $N > N^*$ (upper part of solid black line), and $N_1^* = N - N_2^*$, $N_2^* = \frac{2(\beta + \gamma)}{\gamma} N_1$, when the total travel demand $N < N^*$ (left lower part of solid black line). The red line op shows the occurrence of ($N_1$, $N_2$) when the system achieves UE in the case of $N < N^*$; meanwhile, the red line pq indicates the occurrence of ($N_1$, $N_2$) when the system achieves UE in the case of $N > N^*$. Figure 5a,c and Figure 6e show that the occurrence of early bird parkers increases with the increasing of the parking pricing gap when the parking schedule gap is fixed, which can be shown in Figure 5b,d,f. Figures 4a and 5b show the occurrence of early bird parkers increases with the decreasing of the parking schedule gap when the parking pricing gap is fixed.

### 6.2. The Occurrence Domains of (s, $\Delta t$) When System Achieve UE in Different Cases

This section presents the effect of the parking schedule gap and bottleneck capacity on the allocation of early bird parkers and regular parkers when the commute demand is fixed. Given three cases of total travel demand $N = 550$, $N = 1100$, and $N = 2200$, by varying s and $\Delta t$ we obtain the contour map of the allocation detail between ERPs and RPs in Figure 6. The red, black, and blue dotted lines in Figure 6 show the boundary pattern when $N = N^*$, and the upper part of the dotted line indicates the case of; the domain of ($N_1$, $N_2$) always yields $N_2 = \frac{2(\beta + \gamma)}{\gamma} N_1$ when the system achieves UE, which is not affected by $\Delta t$ and s. By contrast, the bottom part of the dotted line indicates that in the case of $N > N^*$, there will be more commuters who select the early bird parking strategy because the number of

regular parkers will be decided by the domains of $(s, \Delta t)$. Here, the number of $N_2$ always satisfies $N_2 = \frac{\beta + \gamma}{\gamma} \Delta t \cdot s$, and $N_1 = N - N_2$. For example, at the bottom part of dotted line, for the point $A(\Delta t, s) = (10, 9)$, the number of regular parkers is 120, and the number of early bird parkers are $N - N_2$, which are 430, 980 and 2080 when $N = 550$, $N = 1100$ and $N = 2200$, respectively. This indicates that the optimal allocation of early bird parkers and regular parker is decided by the bottleneck capacity and the parking schedule gap when the parking demand (it can also be regarded as morning commuting demand in this paper) is fixed. Otherwise, the allocation contours in Figure 6 can also help parking managers to set the optimal schedule gap or parking supply to maximize the revenue.

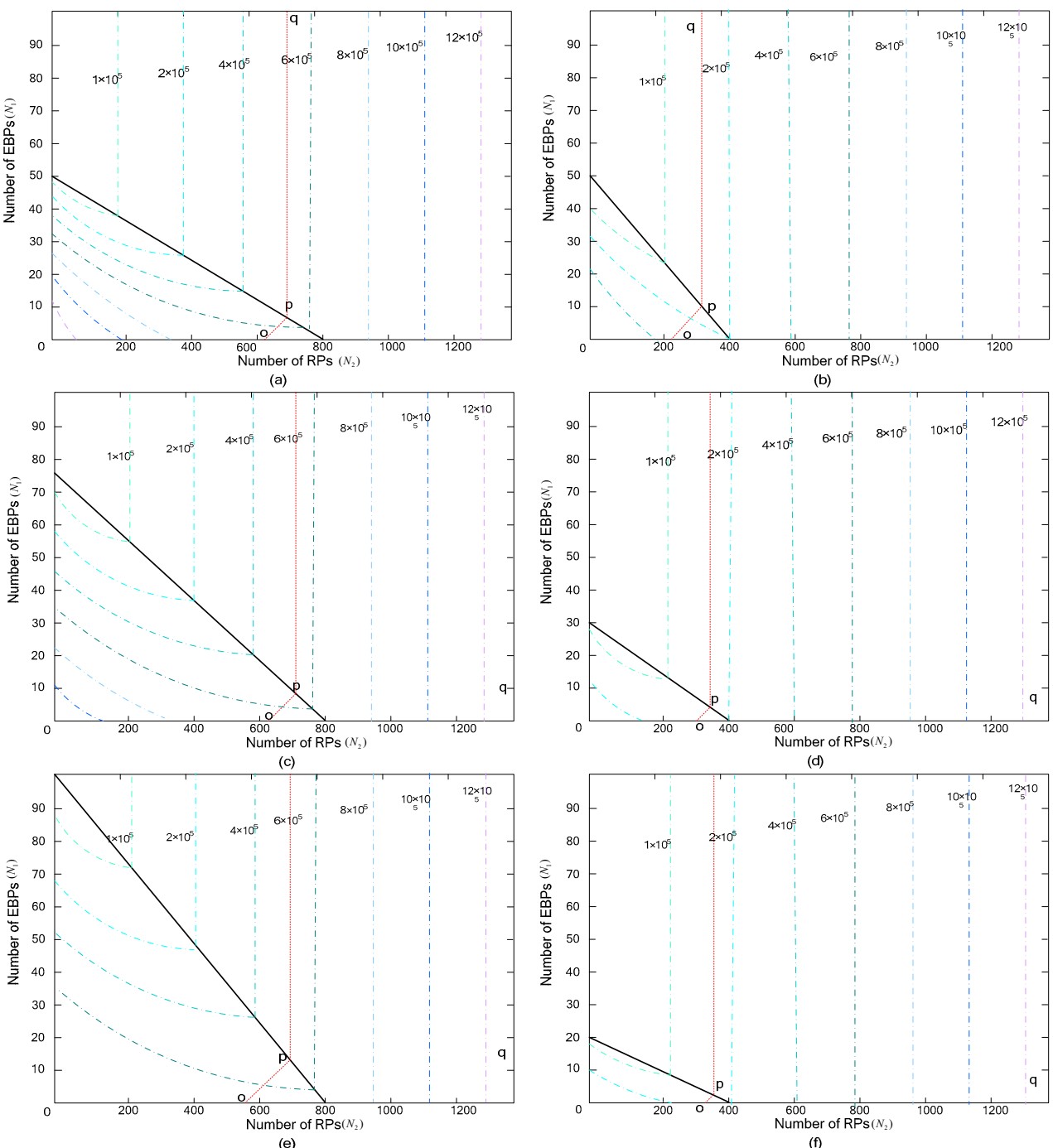

**Figure 5.** The occurrence domains of $(N_1, N_2)$ when the system achieves UE in different cases: (**a**) Case 1 in Table 5; (**b**) Case 2 in Table 5; (**c**) Case 3 in Table 5; (**d**) Case 4 in Table 5; (**e**) Case 5 in Table 5; (**f**) Case 6 in Table 5.

**Table 4.** User equilibrium pattern in different demand cases.

| Cases | Condition | Total Travel Time | Increases in Parking Revenue |
|---|---|---|---|
| $N > N^*$ | $0 < \Delta\tau \leq 2\beta\frac{N}{s} - \frac{2\beta\gamma+\alpha\beta+\alpha\gamma}{\beta+\gamma}\Delta t$ | $\frac{\beta(N-\frac{\beta+\gamma}{\gamma}\cdot\Delta t\cdot s)^2}{\alpha\cdot s} + \frac{\beta+\gamma}{2\gamma}(\Delta t)^2\cdot s$ | $(\frac{\beta+\gamma}{\gamma}\cdot\Delta t\cdot s)\Delta\tau$ |
|  | $2\beta\frac{N}{s} - \frac{2\beta\gamma+\alpha\beta+\alpha\gamma}{\beta+\gamma}\Delta t < \Delta\tau \leq 2\beta\frac{N}{s}$ | $\frac{\beta}{\alpha}\frac{(N_1)^2}{s} + \frac{\gamma}{\beta+\gamma}\frac{(N_2)^2}{2s}$ | $(\frac{2(\beta+\gamma)N}{2\beta+3\gamma} + \frac{\beta+\gamma}{\beta(2\beta+3\gamma)}\cdot\Delta\tau\cdot s)\Delta\tau$ |
|  | $\Delta\tau > 2\beta\frac{N}{s}$ | $\frac{\beta N^2}{\alpha s}$ | $0$ |
| $N \leq N^*$ | $0 < \Delta\tau < 2\beta\frac{N}{s} - \frac{2\beta(\beta+\gamma)}{\gamma}\Delta t$ | $\frac{\gamma}{\beta+\gamma}\frac{N^2}{2s}$ | $(\frac{\gamma}{\beta+\gamma}\frac{N^2}{2s})\Delta\tau$ |
|  | $2\beta\frac{N}{s} - \frac{2\beta(\beta+\gamma)}{\gamma}\Delta t < \Delta\tau \leq 2\beta\frac{N}{s}$ | $\frac{\beta}{\alpha}\frac{(N_1')^2}{s} + \frac{\gamma}{\beta+\gamma}\frac{(N_2')^2}{2s}$ | $(\frac{2(\beta+\gamma)N}{2\beta+3\gamma} + \frac{\beta+\gamma}{\beta(2\beta+3\gamma)}\cdot\Delta\tau\cdot s)\Delta\tau$ |
|  | $\Delta\tau > 2\beta\frac{N}{s}$ | $\frac{\beta N^2}{\alpha s}$ | $0$ |

Where $N_1 = \frac{\gamma}{2\beta+3\gamma}N - \frac{\beta+\gamma}{\beta(2\beta+3\gamma)}\Delta\tau\cdot s$, $N_2 = \frac{2(\beta+\gamma)}{2\beta+3\gamma}N + \frac{\beta+\gamma}{\beta(2\beta+3\gamma)}\Delta\tau\cdot s$, $N_1' = \frac{\gamma}{2\beta+3\gamma}N + \frac{\beta+\gamma}{\beta(2\beta+3\gamma)}\Delta\tau\cdot s - \frac{\beta+\gamma}{2\beta+3\gamma}\Delta t\cdot s$,

$N_2' = \frac{2(\beta+\gamma)}{2\beta+3\gamma}N - \frac{\beta+\gamma}{\beta(2\beta+3\gamma)}\Delta\tau\cdot s + \frac{\beta+\gamma}{2\beta+3\gamma}\Delta t\cdot s$

**Table 5.** User equilibrium pattern in different parameter cases.

| Patern | Schedule Gap between EBPs and RPs | | Parking Fees Gap between EBPs and RPs | | Figure |
|---|---|---|---|---|---|
| Case1 | $t_1^* = 7:00,$ | $t_2^* = 8:00$ | $\tau_1 = 20,$ | $\tau_2 = 10$ | Figure 5a |
| Case2 | $t_1^* = 7:30,$ | $t_2^* = 8:00$ | $\tau_1 = 20,$ | $\tau_2 = 10$ | Figure 5b |
| Case3 | $t_1^* = 7:00,$ | $t_2^* = 8:00$ | $\tau_1 = 20,$ | $\tau_2 = 5$ | Figure 5c |
| Case4 | $t_1^* = 7:30,$ | $t_2^* = 8:00$ | $\tau_1 = 20,$ | $\tau_2 = 14$ | Figure 5d |
| Case5 | $t_1^* = 7:00,$ | $t_2^* = 8:00$ | $\tau_1 = 30,$ | $\tau_2 = 10$ | Figure 5e |
| Case6 | $t_1^* = 7:30,$ | $t_2^* = 8:00$ | $\tau_1 = 20,$ | $\tau_2 = 16$ | Figure 5f |

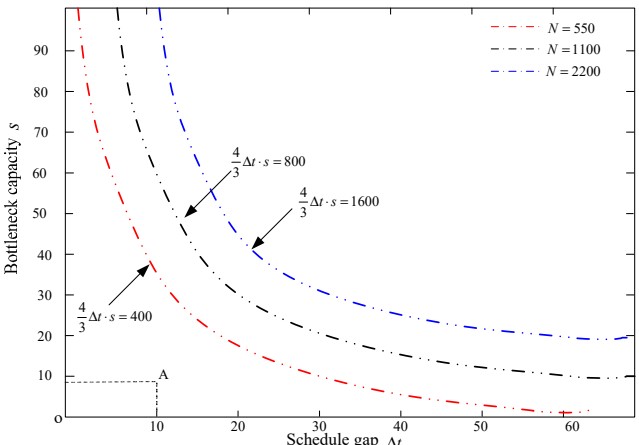

**Figure 6.** The allocation detail between ERPs and RPs the occurrence domains of $(s, \Delta t)$ when system achieve UE.

*6.3. The Occurrence Domains of $(N, \Delta\tau)$ When System Achieve UE in Different Cases*

This section discusses the effect of total commute demand on the total travel time and increases in parking revenue when parking schedule gap and bottleneck capacity is fixed. Given parking schedule gap $\Delta t = 60$ min, and bottleneck capacity s = 10 (veh/min) according to the assumption and some related knowledge in Sections 6.1 and 6.2, it is easy to obtain the optimal parking demand $N^* = 1100$. In order to analyze the effect of parking pricing gap $\Delta\tau$ on the travel time of system, we introduce three cases of $\Delta\tau$ when total travel demand $N > N^*$: $\Delta\tau_1 = 0$ (the red line in Figure 7), $\Delta\tau_2 = \frac{N-1100}{40}$ (the black line in Figure 7) and $\Delta\tau_3 = \frac{N-1100}{20}$ (the blue line in Figure 7), the detail of the relationship between total travel time and parking demands are shown in Figure 7a; meanwhile, the increases in total parking revenue and parking demands are shown in Figure 7b. The number of

regular parkers is 800, and the left is early bird parkers when $\Delta \tau_1 = 0$. We further see that total travel time increases with the parking demand, but the sensitivity of total parking demand for total travel time in different cases of $\Delta \tau$ is different. The increment rate in total travel time decreases with the decrease of $\Delta \tau$. For the increment in parking revenue shown in Figure 7b, the increment in parking revenue $\Delta R$ increases with the varying parking demand, and the increment rate increases with $\Delta \tau$. The system achieves maximum $\Delta R$ and the minimum total travel time when $\Delta \tau = \frac{(N-1100)}{40}$. This means that the scientific arrangement of the parking pricing strategy can help to reduce traffic congestion and increase the revenue of parking managers. These properties is also shown in Figure 8.

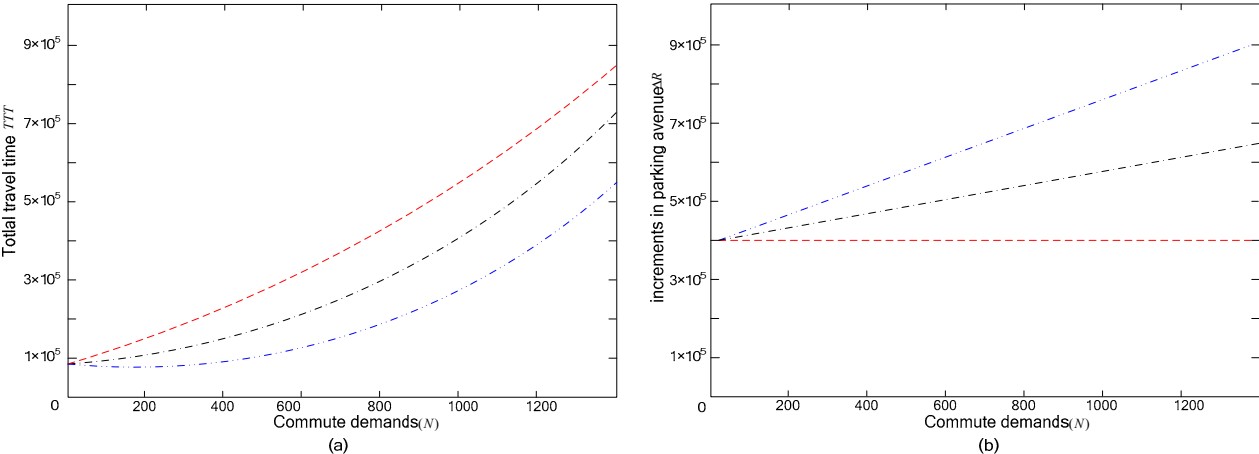

**Figure 7.** The occurrence domains of $(N, \Delta \tau)$ when the system achieves UE: (**a**) the relationship between demand and total travel time; (**b**) the relationship between demand and total parking revenue.

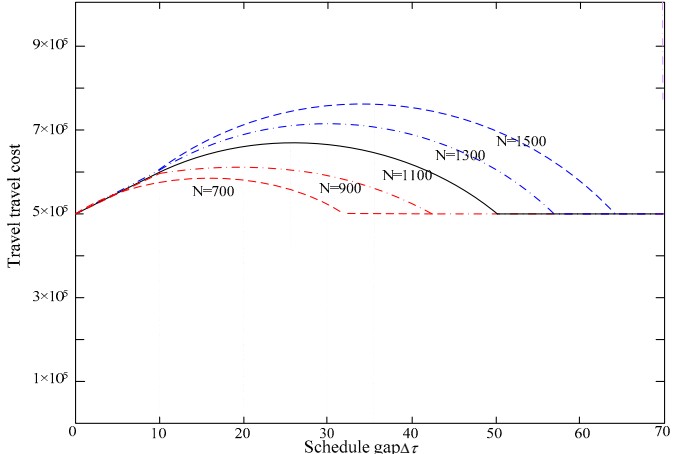

**Figure 8.** The occurrence domains of $(TTT, \Delta \tau)$ when the system achieves UE.

## 7. Conclusions

First of all, this paper confirmed that the traffic congestion has a significant impact on the choice of departure time for morning commuters in consideration of different parking choice strategies between early bird parking and regular parking passengers. When the travel demand or parking demand is relatively small, under the condition of user equilibrium, the impact of commuters from different parking choices on traffic congestion is independent. If the parking prices of EBP and RP are the same, the distribution of these two groups follow the rate $N_2 = \frac{2(\beta+\gamma)}{\gamma} N_1$. When the travel demand is large enough, under the condition of user equilibrium, the distribution of these two groups is affected by the number of regular parkers. On the premise of the smaller parking pricing gap, commuters tend to choose the regular parking schedule. Secondly, this paper revealed that when the

travel demand is different, the sensitivity of the total queuing time and the incremental parking income to the parking pricing gap is different. Therefore, some policies can be taken to adjust the travel demand, reduce the traffic congestion, and increase incremental parking income. Numerical analysis results showed that these above conclusions are consistent with the reality. However, the following shortcomings still exist in this paper. (1) In order to simply reveal the impact of traffic congestion on the departure time choice of morning commuters, significant simplification was made in our model, which is still a certain distance from the actual traffic situation. (2) Parking availability plays a crucial role in traffic congestion modeling and analysis, and the effect of parking space constraints has not been discussed in the process of the early bird mechanism. Therefore, according to the actual situation, further in-depth consideration of departure time preferences, the introduction of flexible travel demand, and restricted parking supply are scientific issues worthy of further discussion in the future.

**Author Contributions:** Writing—original draft preparation, Z.Z.; writing—review, editing and supervision, N.Z. Both authors have read and agreed to the published version of the manuscript.

**Funding:** This research was funded by "Research on the integration of civil aviation and other modes of transportation, grant number 54422601".

**Institutional Review Board Statement:** Not applicable.

**Informed Consent Statement:** Informed consent was obtained from all subjects involved in the study.

**Data Availability Statement:** Not applicable.

**Conflicts of Interest:** The authors declare no conflict of interest.

## Appendix A. The Feature of Commuting Equilibriums

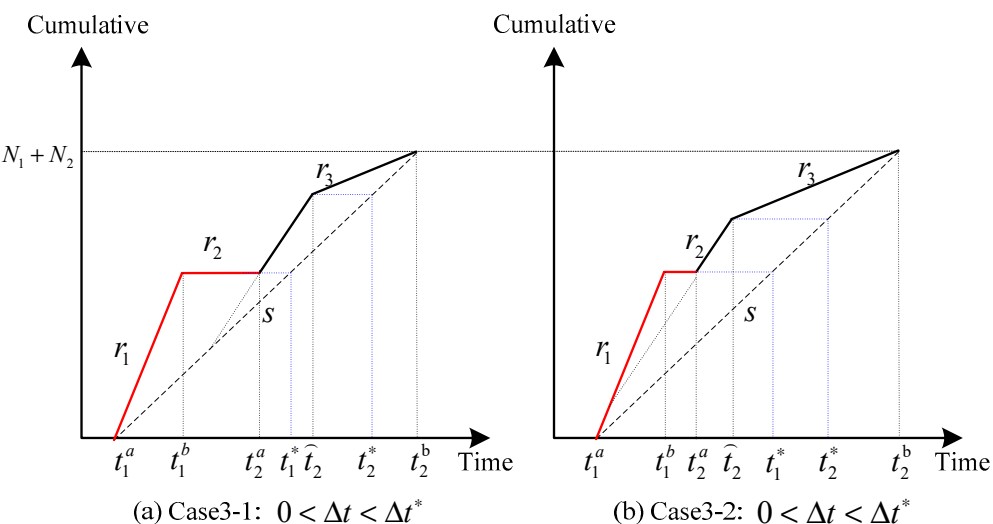

**Figure A1.** All scenarios of commuting equilibriums when all parking spaces are unreserved under a small schedule gap ($0 < \Delta t < \Delta t^*$): (**a**) $\widehat{t}_2$ is earlier than $t_1^*$; (**b**) $\widehat{t}_2$ is larger than $t_1^*$.

## Appendix B. User Equilibrium Pattern When Parking Demand Is Larger Than Standard Case

When parking demand is larger than the standard parking schedule gap ($N > N^*$), according to Theorem 1, under the condition of user equilibrium in the system, we know that the number of early bird parkers and regular parkers is $N_1 = N - N_2$, $N_2 = \frac{\beta + \gamma}{\gamma} \cdot \Delta t \cdot s$. This means that the number of regular parkers is fixed when the parking schedule gap

$\Delta t$ is fixed in the case of $N > N^*$, and it needs more early bird parkers when the parking demand is large. We know that:

$$c_2(t_2^a) = \alpha(t_2^* - t_2^a) + \beta(t_2^* - t_1^*) + \tau_2 = 2\beta \frac{N - \frac{\beta+\gamma}{\gamma} \cdot \Delta t \cdot s}{s} + \beta \cdot \Delta t + \tau_1 \qquad \text{(A1)}$$

$$c_2(t_2^b) = \alpha(t_2^* - t_2^b) + \tau_2 = 2\beta \frac{N - \frac{\beta+\gamma}{\gamma} \cdot \Delta t \cdot s}{s} + \beta \cdot \Delta t + \tau_1 \qquad \text{(A2)}$$

Then we can acquire the earliest and latest departure time of regular parkers:

$$t_2^a = t_2^* - \frac{2\beta}{\alpha} \cdot \frac{N}{s} + \frac{2\beta\gamma}{\alpha(\beta+\gamma)}\Delta t + \frac{1}{\alpha}\Delta\tau \qquad \text{(A3)}$$

$$t_2^b = t_2^* - \frac{2\beta}{\alpha} \cdot \frac{N}{s} + \frac{\beta(\gamma-\beta)}{\alpha(\beta+\gamma)}\Delta t + \frac{1}{\alpha}\Delta\tau \qquad \text{(A4)}$$

$t_2^a$ satisfies $t_1^b \le t_2^a \le t_1^*$, then combining the Equations (A3) and (8):

$$-\alpha\Delta t \le \Delta\tau \le 2\beta\frac{N}{s} - \frac{2\beta\gamma + \alpha\beta + \alpha\gamma}{\beta+\gamma}\Delta t \qquad \text{(A5)}$$

Considering the encouraging early bird parking policy, $\Delta\tau = t_2^* - t_1^* > 0$ then Equation (A5) can be expressed as:

$$0 \le \Delta\tau \le 2\beta\frac{N}{s} - \frac{2\beta\gamma + \alpha\beta + \alpha\gamma}{\beta+\gamma}\Delta t \qquad \text{(A6)}$$

When parking demand is larger than standard parking schedule gap ($N > N^*$), $\Delta\tau$ satisfies Equation (A6), then the system has the same user equilibrium pattern with the case of $\Delta\tau = 0$. The total travel time after UE is $\frac{\beta(N - \frac{\beta+\gamma}{\gamma} \cdot \Delta t \cdot s)^2}{\alpha \cdot s} + \frac{\beta+\gamma}{2\gamma}(\Delta t)^2 \cdot s$ and according the definition of the increasing rate in parking management revenue, $\Delta R = N_2\Delta\tau = (\frac{\beta+\gamma}{\gamma} \cdot \Delta t \cdot s)\Delta\tau$. When $\Delta\tau > 2\beta\frac{N}{s} - \frac{2\beta\gamma + \alpha\beta + \alpha\gamma}{\beta+\gamma}\Delta t$, the travel cost of regular parkers is always more expensive than the travel cost of early bird parkers, and all commuters will select the early bird schedule as the potential travel pattern. No one departs from home after time $t_1^b$ until $\Delta\tau = 2\beta\frac{N}{s}$; here, the cost of EBPs and RPs are equal and these two groups of commuters depart from home independently without a mixed pattern. According to Theorem 3, we can acquire the relationship between EBPs and RPs when the parking schedule is fixed (which can be regarded as zero without loss of generality):

$$N_2 = \frac{2(\beta+\gamma)}{\gamma} \cdot N_1 - \frac{\beta+\gamma}{\beta\gamma}\Delta\tau \cdot s \qquad \text{(A7)}$$

Combing $N = N_1 + N_2$, we can obtain:

$$\begin{cases} N_1 = \frac{\gamma}{2\beta+3\gamma}N - \frac{\beta+\gamma}{\beta(2\beta+3\gamma)}\Delta\tau \cdot s \\[2mm] N_2 = \frac{2(\beta+\gamma)}{2\beta+3\gamma}N + \frac{\beta+\gamma}{\beta(2\beta+3\gamma)}\Delta\tau \cdot s \end{cases} \qquad \text{(A8)}$$

Then the total travel time and increases in parking revenue is $\frac{\beta}{\alpha}\frac{(N_1)^2}{s} + \frac{\gamma}{\beta+\gamma}\frac{(N_2)^2}{2s}$ and $(\frac{2(\beta+\gamma)N}{2\beta+3\gamma} + \frac{\beta+\gamma}{\beta(2\beta+3\gamma)} \cdot \Delta\tau \cdot s)\Delta\tau$, respectively. When $\Delta\tau > 2\beta\frac{N}{s}$, No one departs from home after time $t_1^b$, all commuters select the early bird parking as their potential choice. Then total travel time is $\frac{\beta N^2}{\alpha s}$, and increases in parking revenue $\Delta R = 0$. To summarize, the total

travel time of mixed morning commute system and increases in parking management revenue can be expressed respectively as:

$$
TTT = \begin{cases} \frac{\beta\left(N-\frac{\beta+\gamma}{\gamma}\cdot\Delta t\cdot s\right)^2}{\alpha\cdot s} + \frac{\beta+\gamma}{2\gamma}(\Delta t)^2\cdot s, & 0 \le \Delta\tau \le 2\beta\frac{N}{s} - \frac{2\beta\gamma+\alpha\beta+\alpha\gamma}{\beta+\gamma}\Delta t \\ \frac{\beta}{\alpha}\frac{(N_1)^2}{s} + \frac{\gamma}{\beta+\gamma}\frac{(N_2)^2}{2s}, & 2\beta\frac{N}{s} - \frac{2\beta\gamma+\alpha\beta+\alpha\gamma}{\beta+\gamma}\Delta t < \Delta\tau < 2\beta\frac{N}{s} \\ \frac{\beta N^2}{\alpha s}, & \Delta\tau \ge 2\beta\frac{N}{s} \end{cases}
\tag{A9}
$$

$$
\Delta R = \begin{cases} \left(\frac{\beta+\gamma}{\gamma}\cdot\Delta t\cdot s\right)\Delta\tau, & 0 \le \Delta\tau \le 2\beta\frac{N}{s} - \frac{2\beta\gamma+\alpha\beta+\alpha\gamma}{\beta+\gamma}\Delta t \\ \left(\frac{2(\beta+\gamma)N}{2\beta+3\gamma} + \frac{\beta+\gamma}{\beta(2\beta+3\gamma)}\cdot\Delta\tau\cdot s\right)\Delta\tau, & 2\beta\frac{N}{s} - \frac{2\beta\gamma+\alpha\beta+\alpha\gamma}{\beta+\gamma}\Delta t < \Delta\tau < 2\beta\frac{N}{s} \\ 0, & \Delta\tau \ge 2\beta\frac{N}{s} \end{cases}
\tag{A10}
$$

**Appendix C. User Equilibrium Pattern When Parking Demand Is Smaller Than Standard Case**

According to Theorem 2, when the system achieves user equilibrium at the condition of parking demand less than standard parking schedule gap ($N < N^*$), we can also obtain the relationship between the number of early bird parkers and regular parkers: $N_2' = \frac{2(\beta+\gamma)}{\gamma}\cdot N_1' + \frac{\beta+\gamma}{\gamma}\cdot\Delta t\cdot s - \frac{\beta+\gamma}{\beta\gamma}\Delta\tau\cdot s$. According to the definition of $\Delta R$, the travel cost of early bird parkers and regular parkers respectively is:

$$
c_1(t) = 2\beta\frac{N_1'}{s} + \beta\Delta t, \quad c_2(t) = \frac{\beta\gamma}{\beta+\gamma}\frac{N_2'}{s} + \Delta\tau
\tag{A11}
$$

Combining $c_1(t) = c_2(t)$ and $N = N_1 + N_2$, we obtain the relationship of commuter between EBPs and RPs:

$$
N_1' = \frac{\gamma}{2\beta+3\gamma}N + \frac{\beta+\gamma}{\beta(2\beta+3\gamma)}\Delta\tau\cdot s - \frac{\beta+\gamma}{2\beta+3\gamma}\Delta t\cdot s
\tag{A12}
$$

$$
N_2' = \frac{2(\beta+\gamma)}{2\beta+3\gamma}N - \frac{\beta+\gamma}{\beta(2\beta+3\gamma)}\Delta\tau\cdot s + \frac{\beta+\gamma}{2\beta+3\gamma}\Delta t\cdot s
\tag{A13}
$$

According to theorem 1, when $N < N^*$, $N_2$ yields $N_2' \le \frac{\beta+\gamma}{\gamma}\cdot\Delta t\cdot s$. Then, if we put Equation (A13) into this function, we can get:

$$
\Delta\tau \ge 2\beta\frac{N}{s} - \frac{2\beta(\beta+\gamma)}{\gamma}\cdot\Delta t
\tag{A14}
$$

By using the similarity analysis method with Appendix B, the total travel time of the mixed morning commute system and increases in parking management revenue when $N < N^*$ can be expressed respectively as:

$$
TTT = \begin{cases} \frac{\gamma}{\beta+\gamma}\frac{N^2}{2s}, & 0 < \Delta\tau < 2\beta\frac{N}{s} - \frac{2\beta(\beta+\gamma)}{\gamma}\Delta t \\ \frac{\beta}{\alpha}\frac{(N_1')^2}{s} + \frac{\gamma}{\beta+\gamma}\frac{(N_2')^2}{2s}, & 2\beta\frac{N}{s} - \frac{2\beta(\beta+\gamma)}{\gamma}\Delta t \le \Delta\tau \le 2\beta\frac{N}{s} \\ \frac{\beta N^2}{\alpha s}, & \Delta\tau \ge 2\beta\frac{N}{s} \end{cases}
\tag{A15}
$$

$$
\Delta R = \begin{cases} \left(\frac{\gamma}{\beta+\gamma}\frac{N^2}{2s}\right)\Delta\tau, & 0 < \Delta\tau < 2\beta\frac{N}{s} - \frac{2\beta(\beta+\gamma)}{\gamma}\Delta t \\ \left(\frac{2(\beta+\gamma)}{2\beta+3\gamma}N - \frac{\beta+\gamma}{\beta(2\beta+3\gamma)}\Delta\tau\cdot s + \frac{\beta+\gamma}{2\beta+3\gamma}\Delta t\cdot s\right)\Delta\tau, & 2\beta\frac{N}{s} - \frac{2\beta(\beta+\gamma)}{\gamma}\Delta t < \Delta\tau < 2\beta\frac{N}{s} \\ 0, & \Delta\tau \ge 2\beta\frac{N}{s} \end{cases}
\tag{A16}
$$

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
