# Peer review of "Early Bird Scheme for Parking Management: How Does Parking Play a Role in the Morning Commute Problem"

_sustainability, doi:10.3390/su13158531_

Round 1

Reviewer 1 Report

The topic of this paper is interesting. The methods sound. The results are meaningful and useful. There are several suggestions to improve this paper.

  1. Full stop is missing for many places and sometime "。" is misused.
  2. A map or the simplified network of the studied case is needed.
  3. Major findings of this paper could be summarized in the abstract.

Author Response

Dear Reviewer

I have already revised my paper according to your Comments and Suggestions.

For comments about the several typos, problem in spelling mistakes

The Chinese character of spelling mistakes have been checked all, please see the highlighted part in yellow in revised manuscript, not limit in the problem of “Full stop is missing for many places and sometime "。" is misused”.

For comments about “A map or the simplified network of the studied case is needed.”

I have add the simplified network in the beginning of section 6

The description about a single corridor network are shown in section 3.

 “Considering a bottleneck-constrained corridor that connects a residential area and a central business district (CBD), during the morning peak hours, two groups of continuum homogeneous commuters available on the corridor are early-bird drivers (EBDs), , and standard work drivers (SWDs), . Meanwhile, We assume that all parking spaces are located at the CBD. And the early bird parking service, as a cheaper parking option, can be provided in all park before desired first bird parking time()”

For comments about” Major findings of this paper could be summarized in the abstract.”

 I've rearranged the part of abstract, please see the part of yellow highline

Reviewer 2 Report

I find that the work is analytically correct.
The author has a high specialization in the described problem. I have no objections to the manuscript.

However, there is great doubt as to why the authors avoid quoting the previous manuscript "The morning commute problem with ridesharing when meet stochastic bottleneck" Sustainability 13 (11) 6040. I find that these are two different topics and that the source of the model and general logic is the same.
I emphasize that I conclude the high difference in the results of the manuscript, that the topics are differentiated. Again, a large number of terms are identical from the same authors.
In order to avoid any suspicion of autoplagiarism (I did not find it), I believe that the topics are highly specialized and that this may lead some readers to unnecessary doubt, so I insist that the authors cite their previous manuscript in a suitable way.

Author Response

Dear Reviewer

I have already revised my paper according to your Comments and Suggestions.

For comments about “In order to avoid any suspicion of autoplagiarism (I did not find it), I believe that the topics are highly specialized and that this may lead some readers to unnecessary doubt, so I insist that the authors cite their previous manuscript in a suitable way. “

I've rearranged the part of introduction and add the literature of my previous study in the end of background , please see the part of yellow highlight. To avoid any suspicion of autoplagiarism, we compare the differences between the original literature(The morning commute problem with ridesharing when meet stochastic bottleneck) and this article, especially emphasize the unique contribution for avenue in early bird parking management

Based on the research of our previous study [9] for Vickrey’s bottleneck model, we found that the congestion condition during morning peak hours not only can be effected by commuting demand but also restricted by parking management. Inspired by the previous bi-arrival bottleneck model [9], we focus on the research of management solutions such as early bird parking strategy to optimal the traffic cost for morning commutes under the consideration of the congestion and parking. The novel bottleneck model based on early bird parking mechanism introduced in this paper will include staggering peak parking and differentiated charges, it also can be used to explore the effect on reducing traffic congestion and increasing the parking revenue. Most importantly, the exploration of parking revenue is the biggest contribution in this paper when comparing with the previous model[9]. In this paper ,we also answer the question as below: